# Inhibiting perovskite decomposition by a creeper-inspired strategy enables efficient and stable perovskite solar cells

Shuxian Du[1,2], Hao Huang[1,2], Zhineng Lan[1], Peng Cui [1], Liang Li[1], Min Wang[1], Shujie Qu[1], Luyao Yan[1], Changxu Sun[1], Yingying Yang[1], Xinxin Wang[1] & Meicheng Li [1] ✉

The commercialization of perovskite solar cells is badly limited by stability, an issue determined mainly by perovskite. Herein, inspired by a natural creeper that can cover the walls through suckers, we adopt poly-hexamethyleneguanidine hydrochloride as a molecular creeper on perovskite to inhibit its decomposition starting from the annealing process. The molecule possesses a long-line molecular structure where the guanidinium groups can serve as suckers that strongly anchor cations through multiple hydrogen bonds. These features make the molecular creeper can cover perovskite grains and inhibit perovskite decomposition by suppressing cations' escape. The resulting planar perovskite solar cells achieve an efficiency of 25.42% (certificated 25.36%). Moreover, the perovskite film and device exhibit enhanced stability even under harsh damp-heat conditions. The devices can maintain >96% of their initial efficiency after 1300 hours of operation under 1-sun illumination and 1000 hours of storage under 85% RH, respectively.

Metal halide perovskite solar cells (PSCs) have made impressive progress in photovoltaic performance, showing huge potential in practical commercialization[1–3]. However, a key challenge limiting their commercialization is stability, an issue mainly determined by the perovskite layer[4,5]. At present, organic-inorganic hybrid perovskite film is irreplaceable to achieve high-efficiency (>25%) PSCs, where the main organic component is FAI, and the main inorganic component is $PbI_2$[6,7]. As an organic-inorganic hybrid material, perovskite suffers the instability issues, especially under the conditions of heat, moisture, irradiation, and so on, which urgently need to be addressed[8–11].

Enhancing the stability of perovskite film is crucial for fabricating PSCs with excellent operational stability. As a polycrystalline thin film, the crystallization property and defect states are closely associated with the perovskite stability[12]. Hence, researchers attempt to enhance the perovskite crystallinity and passivate the defects through strategies of crystallization regulation, additive engineering, grain packaging, and so on[13–16]. Recently, it has been reported that vacancy defects can accelerate the absorption of $H_2O$

molecules on perovskite film, initiating the hydration and decomposition of perovskite film in ambient air[17]. Then, a strategy was proposed that blocks hydration by utilizing the guanabenz acetate salt to passivate both cation vacancies and anion vacancies, achieving a high-quality perovskite film with optimized crystallization and enhanced stability in ambient air. In addition, the non-stoichiometry, and composition segregation of the perovskite can also determine their long-term stability[18–20]. For example, the mixed halide perovskite film shows huge potential in fabricating efficient perovskite tandem solar cells, however, the halide segregation severely impacts the film stability[21]. Except from the perspective of halide anion, cation such as $MA^+$ and $FA^+$ also brings difficulty in fabricating perovskite film with excellent stability, since the $MA^+$ and $FA^+$ can be decreased through migration or evaporation under moisture or thermal environment, resulting in the composition segregation and film decomposition[22–24]. Hence, it is expected to be a feasible approach that managing the perovskite components and modifying perovskite film properties to enhance the film stability.

[1]State Key Laboratory of Alternate Electrical Power System with Renewable Energy Sources, School of New Energy, North China Electric Power University, Beijing, China. [2]These authors contributed equally: Shuxian Du, Hao Huang. ✉e-mail: mcli@ncepu.edu.cn

As mentioned above, the film property shows a huge influence on the film stability, which also helps to understand the findings that the perovskite decomposition is an accelerating process[25–27]. It has been revealed that the TiO$_2$-based PSCs show a two-stage ultraviolet degradation, where the decomposition rate of perovskite film is accelerating gradually[28]. Schelling's segregation model has been used to study individual cation migration and phase segregation, and it is found that the perovskite decomposition rate is accelerating and the initial film inhomogeneity can accelerate materials degradation[29]. As for the solution-prepared perovskite film, the annealing process is crucial for perovskite crystallization. However, the heat condition also has detrimental impacts on the as-formed perovskite film, since the organic component is ready to evaporate. So, it is necessary to comprehensively evaluate the influence of annealing on the perovskite property, including its stability, and further propose a strategy that can inhibit perovskite decomposition starting from the annealing process, leading to an ideal perovskite film with both excellent photoelectric properties and stability.

In this work, we emphasize the dynamic stability of perovskite film during the annealing process and reveal that the perovskite film exhibits an obvious decomposition as the annealing proceeds, resulting in an initial decomposition state. Inspired by the natural creeper which can steadily cover the walls through suckers, poly-hexamethyleneguanidine hydrochloride (PHMG) has been utilized to serve as a molecular creeper to inhibit the perovskite decomposition starting from the annealing process. PHMG possesses a long-line molecular structure and well-distributed guanidine groups that can interact with FA$^+$ through N···H bond, making the PHMG be absorbed strongly on perovskite. The PHMG can optimize perovskite crystallization, and inhibit the FA$^+$ migration and escape away even under high temperature, leading to a high-quality perovskite film with excellent photoelectric properties and stability. The resulting PSCs achieve a PCE of 25.42% (certificated 25.36%). Moreover, the PSCs can maintain >96% of their initial PCE after 1300-h of operation under 1-sun illumination and 1000-h of storage under 85% RH, respectively.

## Results

### Creeper-inspired design and interaction mechanism

Human social life is always closely related to natural plants. Among them, the creeper plant is familiar to us since it often emerges on the wall, such as the Boston ivy. Creepers can provide a reliable cover for the naked walls, which not only decorates the wall but also inhibits the components of the wall from escaping, even on rainy days with strong wind. Inspired by the creeper, we try to construct a molecular creeper on perovskite to address the stability issue. The perovskite decomposition can be simply divided into two steps: the first step is the organic cation escape, and the second step is the inorganic framework collapse[30]. Considering that the organic component is relatively more unstable than the inorganic component in the hybrid perovskite material, we attempt to construct a molecular creeper that can anchor the organic cation using its suckers and further cover the perovskite (Supplementary Fig. 1). Such a construction of molecular creeper is expected to show a positive effect on perovskite stability. The PHMG is a polymer that possesses a long-line molecular structure with well-distributed guanidine groups. The guanidine group can strongly interact with FA$^+$ through multiple N···H bonds, which makes it steadily cover the perovskite grain, just like the creepers covering the walls (Fig. 1a). Hence, the PHMG is rationally selected to serve as the molecular creeper.

To explore the reasonable configuration of PHMG on perovskite, the density functional theory (DFT) calculation was utilized. Notably, during the DFT calculation, we introduced the single unit of PHMG on perovskite. As shown in Fig. 1b, the optimized geometry exhibits that the PHMG can be absorbed on perovskite through hydrogen bonds between the FA$^+$ and guanidine group. The absorption energy of the

PHMG unit on perovskite is calculated to be −3.47 eV (Fig. 1c), which demonstrates that the PHMG can strongly cover the perovskite and further immobilize the FA$^+$. We also calculated the formation energy of FA$^+$ vacancy (Fig. 1c). After introducing the PHMG unit, the formation energy of FA$^+$ vacancy increases from 1.29 eV to 1.94 eV, which also validates that the PHMG can effectively immobilize the FA$^+$.

To certify the interaction between PHMG and perovskite, experimental measurements such as the $^1$H nuclear magnetic resonance ($^1$H NMR), Fourier transform infrared spectroscopy (FTIR), and Raman spectra were carried out. In the $^1$H NMR spectra shown in Fig. 1d, the characteristic H peak of 8.71 ppm belongs to the FA$^+$ shifts to 8.74 ppm, which should be due to the interaction with PHMG. The signal corresponding to the interaction between FAI and PHMG has also been captured by FTIR spectra, where the N-H stretching vibration peak in FAI exhibits a downward shift from 3341 to 3312 cm$^{-1}$ (Fig. 1e), indicating the formation of hydrogen bonding between FAI and PHMG. The Raman spectra also suggest the existing hydrogen bond between FAI and PHMG, where the characteristic N-H stretching vibration of FA$^+$ located at 3198 cm$^{-1}$ shifts toward higher wavenumbers (Fig. 1f). Thus, we confirm that the PHMG can be absorbed on perovskite through hydrogen bond between PHMG and FA$^+$, which contributes to anchoring the FA$^+$ and inhibiting it from escaping away.

### Perovskite crystallization and decomposition inhibition at the annealing process

After assuring the PHMG can cover perovskite steadily, just like a creeper, we further systematically explore the influence of PHMG on perovskite properties, especially during the annealing process. The perovskite films with different annealing times were characterized by measurements of scanning electron microscope (SEM) and X-ray diffraction (XRD). As shown in Fig. 2a and Supplementary Fig. 2, after annealing for 3 min, both the control and target films show complete perovskite grains and continuous grain boundary. At this stage, we can notice that there is unreacted PbI$_2$ located at the boundary in the control films. As a comparison, there is no PbI$_2$ in the target films, which suggests that the introduced PHMG can promote the reaction between PbI$_2$ and organic halide. Notably, this promoted reaction can be well demonstrated by the in situ spectra measurement that will be comprehensively discussed in the following. As the annealing time gradually increases to 15 min, we can observe the apparent increment of PbI$_2$ in the control film. The increased PbI$_2$ is mainly located at the boundary, which suggests that the PbI$_2$ comes from the perovskite decomposition due to FA$^+$ evaporation under high temperature. In contrast, the PbI$_2$ in the target films is hardly increased and we can still observe the clear and complete perovskite grain and boundary, demonstrating that the PHMG can effectively inhibit the escape of FA$^+$ and further inhibit perovskite decomposition under high temperature. The results of atomic force microscopy (AFM) and Kelvin probe force microscopy (KPFM) also validate that the perovskite undergoes decomposition accompanied by generation of PbI$_2$ in the annealing process, and the creeper-inspired strategy can effectively inhibit this decomposition (Supplementary Fig. 3 and Supplementary Note 1).

The full XRD spectra of perovskite films with different annealing times exhibited in Supplementary Fig. 4 also synchronously show the phenomenon of PbI$_2$ increase for the control film. We extracted the data of the PbI$_2$ peak and (001) peak from the XRD spectra and made a detailed comparison. Figure 2b, c exhibit the evolution of PbI$_2$ peak intensity and peak ratio of PbI$_2$/(001). In the control film, the intensity of PbI$_2$ peak continuously increases as the annealing time increases, and the peak ratio of PbI$_2$/(001) also shows a similar tendency to keep increasing. As a comparison, in the target film, the intensity of the PbI$_2$ peak only possesses a little increment with the annealing time increasing, and the peak ratio of PbI$_2$/(001) shows a minor change. These results of XRD demonstrate that the perovskite decomposes

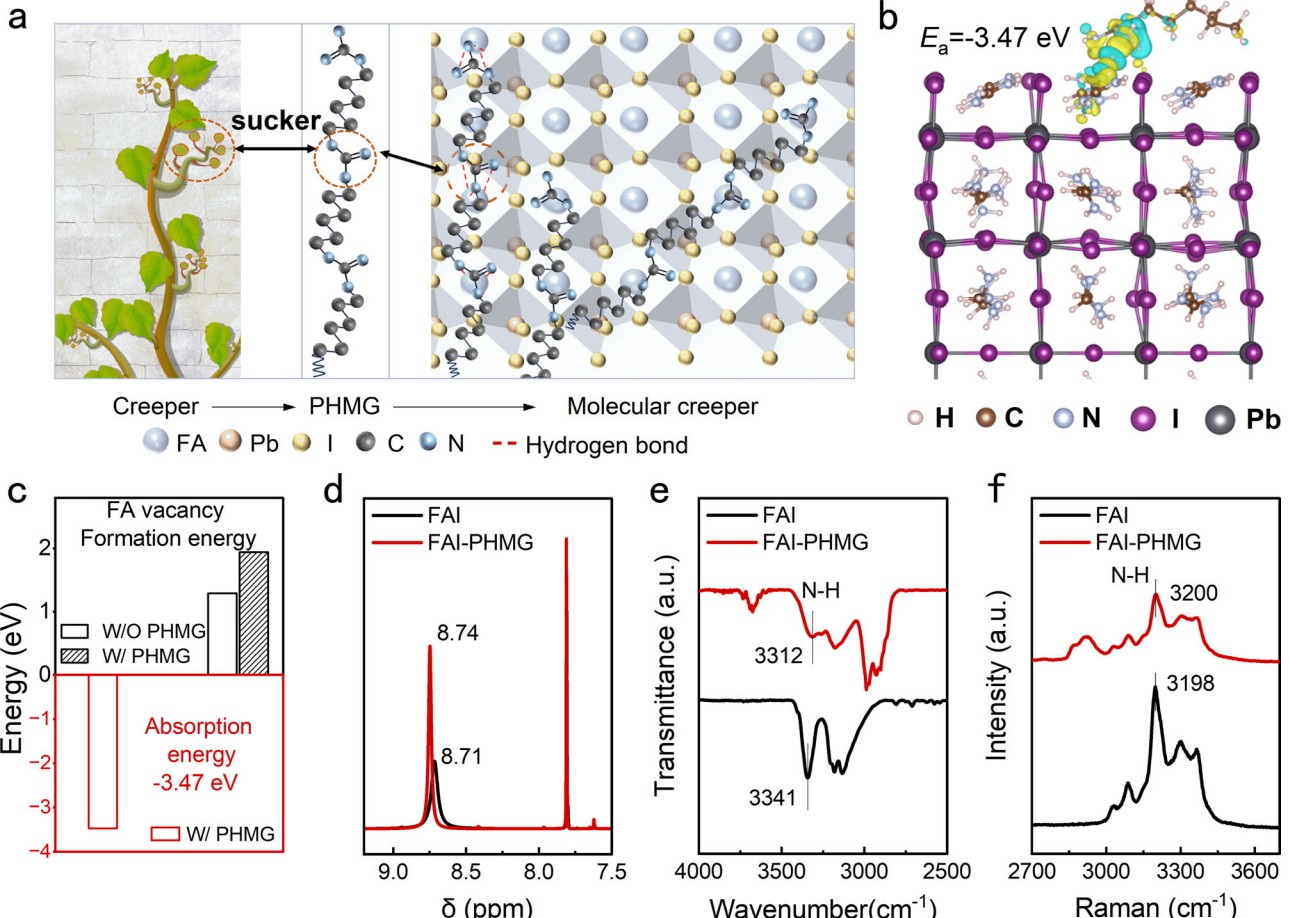

**Fig. 1 | Creeper-inspired protection strategy achieved by PHMG. a** Schematic diagram of Creeper-inspired protection strategy. **b** The charge density difference of the single unit of PHMG anchoring on the perovskite grain surface. **c** The adsorption energy of the single unit of PHMG on perovskite and FA⁺ vacancy formation energy with and without the single unit of PHMG. **d** $^1$H NMR of the FAI and FAI-PHMG dissolved in DMSO-d6. **e** FTIR spectra and (**f**) Raman spectra of the FAI and FAI-PHMG ground powers.

during the annealing process. This decomposition mainly results from the escape away of FA⁺ under heating conditions and generates a great deal of PbI$_2$. Although PbI$_2$ located at grain boundary can passivate defects according to the previous reports, PbI$_2$ also brings a detrimental impact on perovskite stability, especially under light illumination[31–37]. We utilized PHMG to serve as a molecular creeper on perovskite, which can effectively suppress the decomposition during the annealing process by anchoring the FA⁺ and covering the grains. The positive effect of molecular creeper on inhibiting perovskite decomposition during the annealing process is also validated by the X-ray photoelectron spectroscopy (XPS) measurements. As shown in Fig. 2d, the Pb⁰ peaks mainly generated from the perovskite decomposition have been inhibited by utilizing PHMG.

To directly study the influence of PHMG on perovskite crystallization, we performed in situ UV visible absorption (UV-vis) and in situ Photoluminescence (PL) to investigate perovskite crystallization during spin-coating in the N$_2$ glovebox and annealing process in the air environment, respectively. As for the spin-coating process after dropping the organic halide solution on PbI$_2$ film, the UV-vis spectra for both films (Fig. 3a, b) show similar features with a quick intensity climbing stage at first 5 s. This quick-intensity climbing stage may result from the reaction between PbI$_2$ and organic halide. It is noted that the quick intensity climbing stage in the target film is earlier for 2 s than that in the control film, which demonstrates that the PHMG can promote the reaction between PbI$_2$ and organic halide (Supplementary Fig. 5 and Supplementary Note 2). This quick reaction can also be validated by the in situ PL spectra in Fig. 3c, d. Both the spectra of the control film and target film show a sharp intensity rise followed by a sharp quenching, which should be associated with the reaction between PbI$_2$ and the organic halide and the throwing off of the solution as the spin-coating begins. In detail, the stage of sharp intensity changes in the target film emerged earlier for ~2.5 s than that in the control film, which is consistent with the results of in situ UV-vis spectra. The peak of PL spectra for the spin-coating process is located at ~780 nm, indicating the formation of α-phase FAPbI$_3$. At this stage, the formed FAPbI$_3$ may be the initial nucleus for the following crystal growth. Hence, the introduced PHMG can promote the reaction between PbI$_2$ and organic halide, leading to an accelerated nucleation.

During the annealing process, we can deconvolute the crystallization into three stages combining the results of in situ UV-vis and PL[38–41]. Stage I corresponds to the fast crystallization which can be evidenced by the rapid intensity increase at UV-vis and PL spectra. Notably, stage I should coexist the nucleation and crystal growth since the film begins to be heated. The following sharp intensity decrease should be due to the solvent volatilization which makes the wet film transfer to solid film. The phase transition from liquid to solid may have an enormous influence on the results of spectra measurements. The crystallization in stage I of both the control and target films is almost the same, which can be explained by that the quick applied high temperature dominates the crystallization in this stage. In stage II, the crystallization that should correspond to the crystal restructuring of both the control and target films begins to show obvious differences

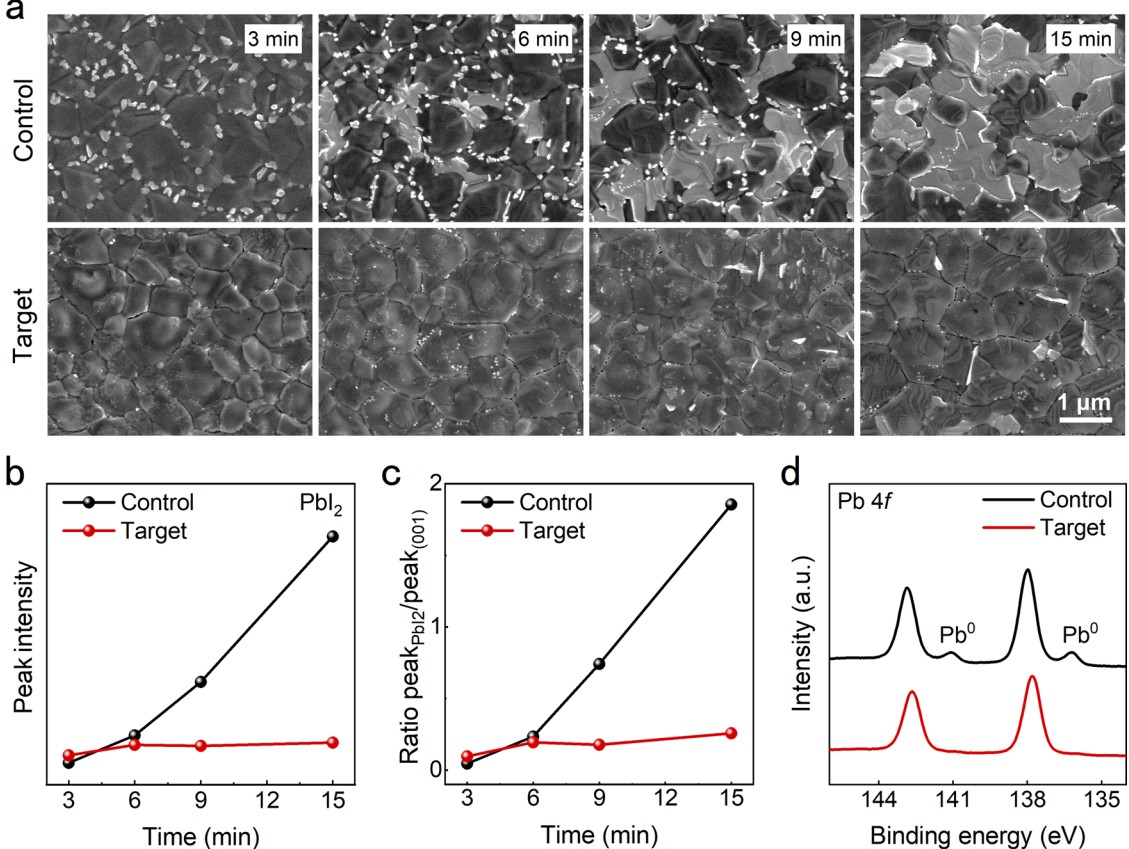

**Fig. 2 | Inhibited perovskite decomposition during the annealing process.**
**a** Surface SEM images of control and target films with different annealing times.
**b** Peak intensity of $PbI_2$ of control and target films with different annealing times. **c** Peak intensity ratio of $PbI_2$/(001) of control and target films with different annealing times. **d** Pb 4f spectra of control and target films.

on a time scale. The longer time in stage II suggests that the crystal restructuring in the target film is slower and more complete than that in the control film. Then, the crystallization process comes into stage III, a stage corresponding to crystallization enhancement accompanied by decomposition. In this stage, the absorption intensity and PL intensity show a tendency to slowly increase for both the control and target films, indicating the crystallization enhancement. Since the $PbI_2$ possesses the ability to passivate defects, the decomposition is not exhibited obviously in these spectra results. Based on this in situ characterization, the PHMG, which served as the molecular creeper, can accelerate the nucleation and delay the crystal growth, which is beneficial for improving the property of perovskite film.

**Photovoltaic performance of PSCs**

To gain an insight into the influence of PHMG on the photoelectric properties and defect states of perovskite films, we performed PL mapping, time-resolved photoluminescence (TRPL) spectra, electroluminescence (EL), and photoluminescence quantum efficiency (PLQE) on control and target films. As shown in Fig. 4a, the target film has a higher fluorescence intensity than the control film. In addition, the carrier lifetime of the target films prolongs from 775 ns to 1148 ns (Fig. 4b). Both the results of PL mapping and TRPL confirm that introducing PHMG can effectively reduce the defects of perovskite film, leading to a suppressed non-radiative recombination which is also validated by the increased EL intensity shown in (Supplementary Fig. 6). The PLQE results display that the PHMG led to increasing luminous efficiencies from 1.81% to 13.06% (Supplementary Fig. 7), which also indicate that there is suppressed non-radiative recombination in the target film. We further quantitatively evaluated the defect density of both films by testing the space charge-limited current

(SCLC). After introducing PHMG into the electron-only device and the hole-only device, the trap density ($N_{trap}$) decreases from $10.67 \times 10^{15}$ to $8.03 \times 10^{15}$ cm$^{-3}$ and from $2.05 \times 10^{15}$ to $1.53 \times 10^{15}$ cm$^{-3}$, respectively (Fig. 4c, Supplementary Figs. 8 and 9 and Supplementary Note 3). The reduced defects in perovskite film can contribute to suppressing non-radiative recombination, which is beneficial to enhancing the photovoltaic performance of PSCs.

We fabricated planar PSCs with the structure of FTO/TiO$_2$/perovskite/Spiro-OMeTAD/Au to explore the influence of PHMG on their photovoltaic performance. After determining the optimal concentration of PHMG (Supplementary Fig. 10), we collected the photovoltaic parameters of 40 control and target PSCs, respectively. The PCE distribution histograms indicate that both the PSCs possess good reproducibility, and the target PSCs possess a higher average PCE of 24.84% than that (23.80%) of control PSCs (Fig. 4d). The corresponding distributions of open circuit voltage ($V_{OC}$), short circuit current density ($J_{SC}$), fill factor (FF) are shown in Supplementary Fig. 11. The target PSCs achieve a champion PCE of 25.42% ($J_{SC} = 26.20$ mA cm$^{-2}$, $V_{OC} = 1.161$ V, FF = 83.53%) with a minor hysteresis (Fig. 4e). The corresponding external quantum efficiency spectrum shown in Supplementary Fig. 12 yields an integrated $J_{SC}$ of 25.72 mA cm$^{-2}$, showing a small variation from the value obtained from $J$-$V$ measurements. Impressively, the PCE of 25.42% is the highest value among the TiO$_2$-based planar PSCs reported to date (Fig. 4f and Supplementary Table 1). Moreover, one of our best-performing PSCs has been sent to a third-independent institute of the National Institute of Metrology for certification and obtained a certificated PCE of 25.36% (Supplementary Fig. 13). In contrast, the control PSCs achieved a champion PCE of 24.29% ($J_{SC} = 26.15$ mA cm$^{-2}$, $V_{OC} = 1.134$ V, FF = 81.88%). It is noted that the

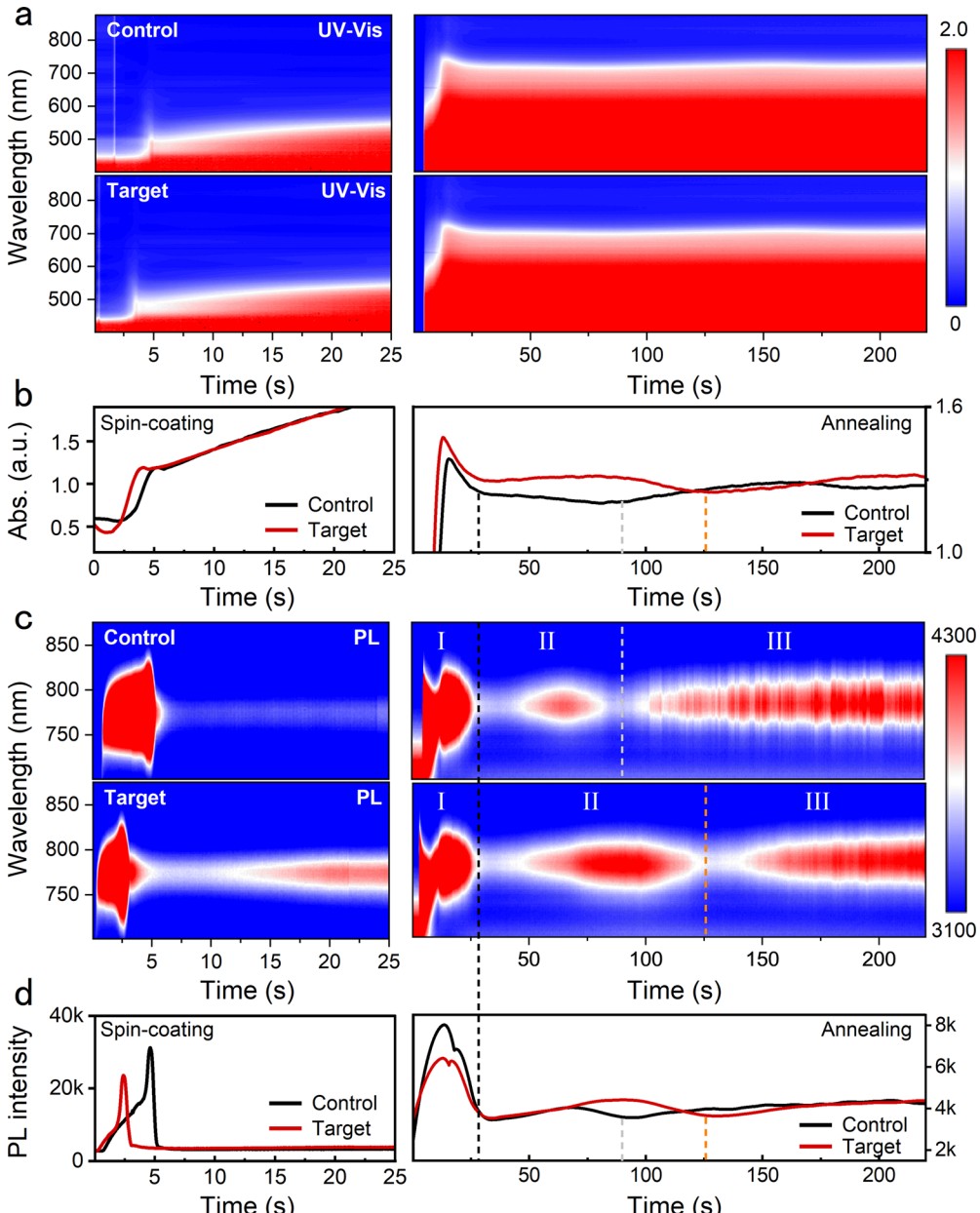

**Fig. 3 | In situ characterization of perovskite crystallization. a, b** In situ UV-vis spectra of control and target perovskite films during spin-coating. **c, d** In situ PL spectra of control and target perovskite films during annealing.

increment of PCE mainly results from the increase of $V_{OC}$, which should be ascribed to the suppressed non-radiative recombination. The suppressed non-radiative recombination has also been validated by the measurement of dark $J$-$V$ curves and ideality factor (Supplementary Figs. 14 and 15).

In this work, if not specified, the perovskite film is fabricated by using a two-step spin-coating method. To demonstrate the reliable effect of this creeper-inspired strategy on improving the photovoltaic performance of PSCs, we further fabricated perovskite film using one-step spin-coating method, and the PHMG was introduced into the perovskite precursor. The results are shown in Fig. 4g–i. As shown in Fig. 4g, the PCE distribution histograms indicate that the target PSCs exhibit an average PCE of 24.98%, which is higher than that (23.71%) of the control PSCs. In Fig. 4h, the target PSCs achieve a champion PCE of 25.41% ($J_{SC}$ = 25.95 mA cm$^{-2}$, $V_{OC}$ = 1.174 V, FF = 83.43%). As a comparison, the control PSCs achieve a champion PCE of 24.57% ($J_{SC}$ = 25.95 mA cm$^{-2}$, $V_{OC}$ = 1.147 V, FF = 82.54%). The PCE increment

demonstrates that the PHMG shows a reliable effect on improving the photovoltaic performance of PSCs and possesses applicability in perovskite films fabricated by different methods. In addition, the stabilized power output for 300 s of both PSCs was measured. The PCE of target PSCs is 25.42% with negligible decay after tracking for 300 s at maximum power output point, while the PCE of control PSCs decays from 24.39% to 23.41% (Fig. 4i).

The results of stabilized power output for 300 s demonstrate that the target PSCs show enhanced stability, which should result from the optimized perovskite films. The molecular creeper of PHMG can anchor FA$^+$ strongly and cover perovskite grains, which is expected to mitigate the ion migration, an intrinsic factor that badly impacts the device stability[42,43]. To investigate the ion migration, we designed a novel characterization by combining the measurement of PL and electrochemical workstation to reveal the hysteretic PL intensity responses when changing the external electric field, the PL-V hysteresis (Fig. 5a). The external electric field ($E_{ex}$) has multiple

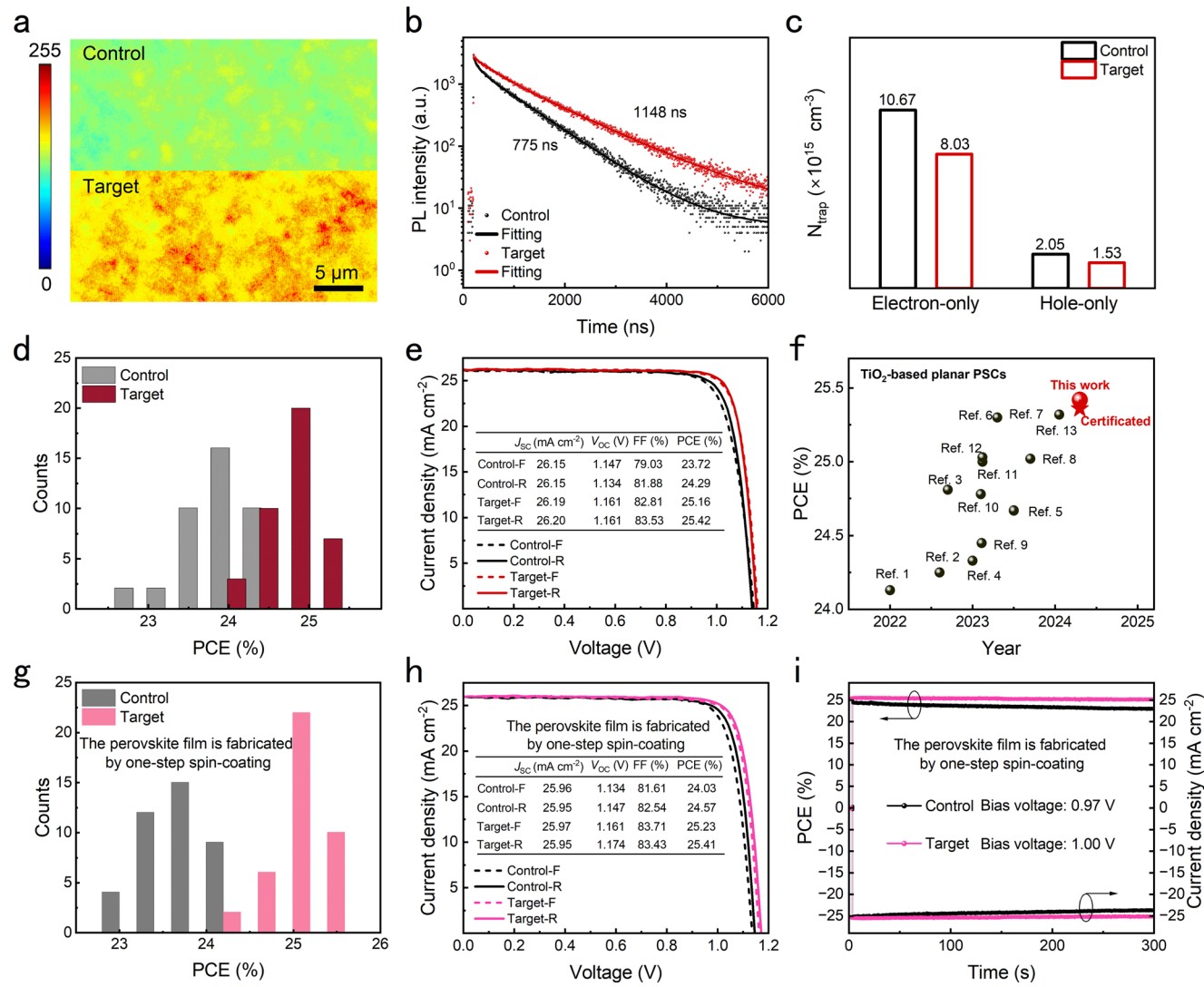

**Fig. 4 | Photoelectric performance of perovskite films and PSCs. a** PL mapping and (**b**) TRPL spectra of control and target films. **c** The defect density of control and target films was measured by using SCLC measurement. **d** Distribution histograms of PCE value among 40 control and target PSCs, respectively. Source data are provided as a Source Data file. **e** J-V curves (reverse and forward scan) of control and target PSCs with an active area of $0.08 cm^2$. **f** PCE comparison of our PSCs with the reported efficient PSCs that apply TiO$_2$ as the electron transport layer. **g** Distribution histograms of PCE value among 40 control and target PSCs, respectively. Source data are provided as a Source Data file. **h** J-V curves (reverse and forward scan) of control and target PSCs with an active area of 0.08 cm$^2$. **i** Stabilized power output of control and target PSCs. Notably, in Fig. 4g–i, the perovskite films in the PSCs are fabricated by one-step spin-coating.

effects on the photoexcited states of PSCs. Firstly, the $E_{ex}$ will instantaneously change the drift velocity of photogenerated charge carriers by influencing the practical electric field within the device. Secondly, the $E_{ex}$ will cause ion migration, which can change the distribution and density of trap states that are associated with non-radiative recombination. As shown in Fig. 5b, c, compared to control PSCs, the target PSCs show smaller PL-V hysteresis. Since the instant PL response on the $E_{ex}$ is supposed not to be responsible for PL-V hysteresis, the ion migration that exhibits relatively slow kinetics should be the reason. So, the reduced PL-V hysteresis indicates that the ion migration in target PSCs has been effectively suppressed. To further clarify the type of mobile ions, the measurement of time-of-flight secondary ion mass spectrometry (TOF-SIMS) was carried out on the control and target PSCs that stored in ambient air for about 900 h without any other accelerating treatment (Fig. 5d). The results of TOF-SIMS indicate that the PHMG can effectively inhibit the FA$^+$ migration, which is consistent to our analysis that the PHMG can anchor FA$^+$ strongly. The mobile ion concentrations in the control

and target PSCs were calculated by transient ion-migration currents measurement[44,45]. It is found that the average mobile ion concentration of target PSCs is $6.07 \times 10^{16} cm^{-3}$, which is lower than that ($17.28 \times 10^{16} cm^{-3}$) of control PSCs (Fig. 5e, Supplementary Table 2, and Supplementary Note 4). In addition, the open-circuit voltage decay (OCVD) characteristics of the PSCs were measured to evaluate charge reorganization associated with ion mobility (Fig. 5f)[46,47]. The fast decay can be ascribed to the free charge-induced recombination, while the slow decay is attributed to the ion accumulation-related charge recombination. It is noted that the photovoltage decay of target PSCs in a slow decay process is slower than control PSCs, indicating that the ion migration in target PSCs is suppressed by PHMG. The effectiveness of the creeper-inspired strategy on inhibiting ion migration has been also validated by the decreased mobile ion migration in both PSCs based on MAPbI$_3$ and MA$_{0.75}$FA$_{0.25}$PbI$_3$, indicating that this strategy can also be well applied in perovskite with different component (Supplementary Fig. 16 and Supplementary Tables 3 and 4).

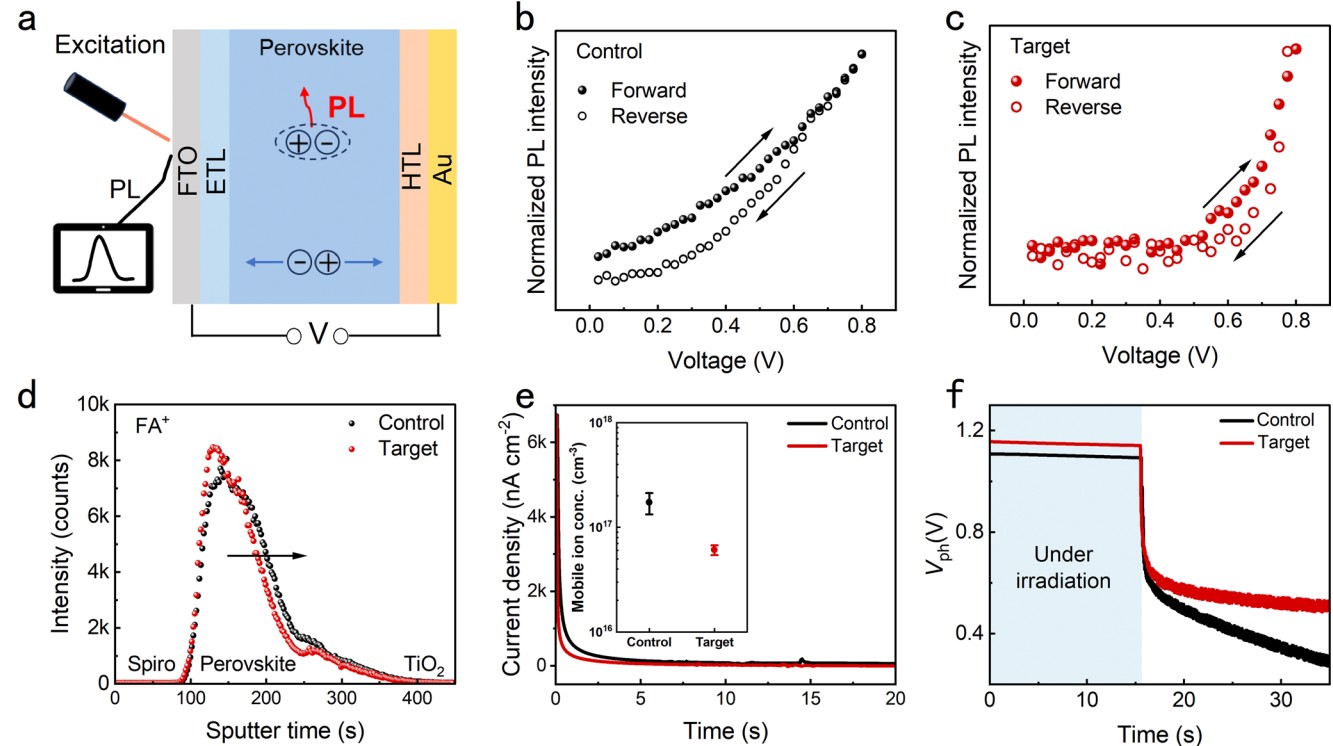

**Fig. 5 | Characterization of ion migration. a** Schematic diagram of PL-V hysteresis test. The PL-V curves for forward and reverse scans between 0 and +0.8 V of (**b**) control and (**c**) target PSCs. **d** TOF-SIMS of control and target PSCs after storage in ambient air for about 900 h without any other accelerating treatment. **e** Transient ion-migration currents of PSCs and mobile ion concentrations in the control and target PSCs, 1.1 V forward bias applied to devices is removed. **f** OCVD curves of control and target PSCs after light soaking under AM1.5 illumination for 15 s. ($V_{ph}$ stands for photovoltage and the blue area represents that the PSCs are under irradiation).

## Stability of perovskite film and devcie

The introduced PHMG can strongly anchor the FA$^+$ and cover the grain like a creeper, showing a positive effect on suppressing ion migration and inhibiting perovskite decomposition starting from the annealing process. The inhibited decomposition at the initial stage is also beneficial to enhance the perovskite stability after the film is applied in the PSCs. To systematically evaluate the stability of perovskite film, we carried out a series of tests through accelerating aging under harsh conditions. We placed both films under three different harsh conditions (Supplementary Fig. 17), 85% RH at room temperature, 85°C heating in ambient air, and 1-sun illumination in ambient air, and then monitored the film changes by measurement of SEM and XRD. As shown in Fig. 6a and Supplementary Fig. 18, we keep observing perovskite films aged under a condition of 85% RH at room temperature for 96 h. As the aging time increases, the control film shows an obvious decomposition, evidenced by the increase of PbI$_2$ and the emergence of a broken structure. Besides, the emergence of δ-FAPbI$_3$ in the control film after aging for 24 h keeps increasing with the aging time prolonging, which can also demonstrate that the control film shows decomposition under the high-humidity condition. In contrast, both the results of SEM and XRD show that the target film can maintain the initial state even under the harsh condition of 85% RH for 96 h, proving that the target film possesses enhanced stability after we rule out the factor of contact angle (Supplementary Fig. 19). High temperature is also an unavoidable factor since the practical temperature of PSCs under a routine operation can reach ~65 °C[48,49]. We explored the enhancement effect of PHMG on the thermal stability of perovskite film by placing films in a condition of 85°C in ambient air (~25% RH). In Fig. 6b left, as for the target film, there are almost no changes on peaks (001) and (111), and only a slight increase in the peak intensity of PbI$_2$ after heating for 100 h. As a comparison, the peak intensity of control (001) and (111) exhibits a dramatic decrease after heating for 100 h

accompanied by the PbI$_2$ peak enhancement. These results prove that the PHMG can effectively enhance the thermal stability of perovskite film. In our work, the PHMG can strongly anchor the FA$^+$ through multiple hydrogen bonds and further cover the grain steadily. Such protection can not only prevent the combination between H$_2$O and FA$^+$ but also inhibit the FA$^+$ from escaping away, inhibiting perovskite decomposition even under the harsh conditions of high humidity and high temperature.

The molecular creeper of PHMG can inhibit perovskite decomposition from the annealing process, suppressing the excess PbI$_2$ at the boundary. PbI$_2$ is prone to decompose into metallic lead Pb$^0$ and gaseous I$_2$ under continuous light, deteriorating the photo-instability of perovskite film[50,51]. Considering the detrimental impact of PbI$_2$, we further probe the light-soaking stability of both perovskite films by placing them under 1-sun illumination for 100 h in an ambient environment. Notably, As shown in Fig. 6b right, the PbI$_2$ peak became the strongest peak instead of the original peak (111) in the control film while the target film maintains the initial XRD spectrum basically, suggesting that PHMG can improve the illumination tolerance of perovskite. Notably, there should be two reasons for the enhanced stability of perovskite film. The first one is the PHMG can strongly anchor the FA$^+$ through multiple hydrogen bonds and further cover the perovskite grain steadily, which inhibits the FA$^+$ migration and escape away. The second one is that the PHMG inhibits the perovskite decomposition as early as the annealing process, which suppresses the excess PbI$_2$ at the boundary and eliminates the detrimental effect of PbI$_2$ on perovskite stability, especially under light and heat conditions.

After discussing the film stability and ion migration, we further characterize the device stability. Firstly, the long-term storage stability was conducted with both unencapsulated devices placed in ambient air. The unencapsulated target device can maintain ~96% of its initial PCE after aging for 2000 h, while the control device decays badly after

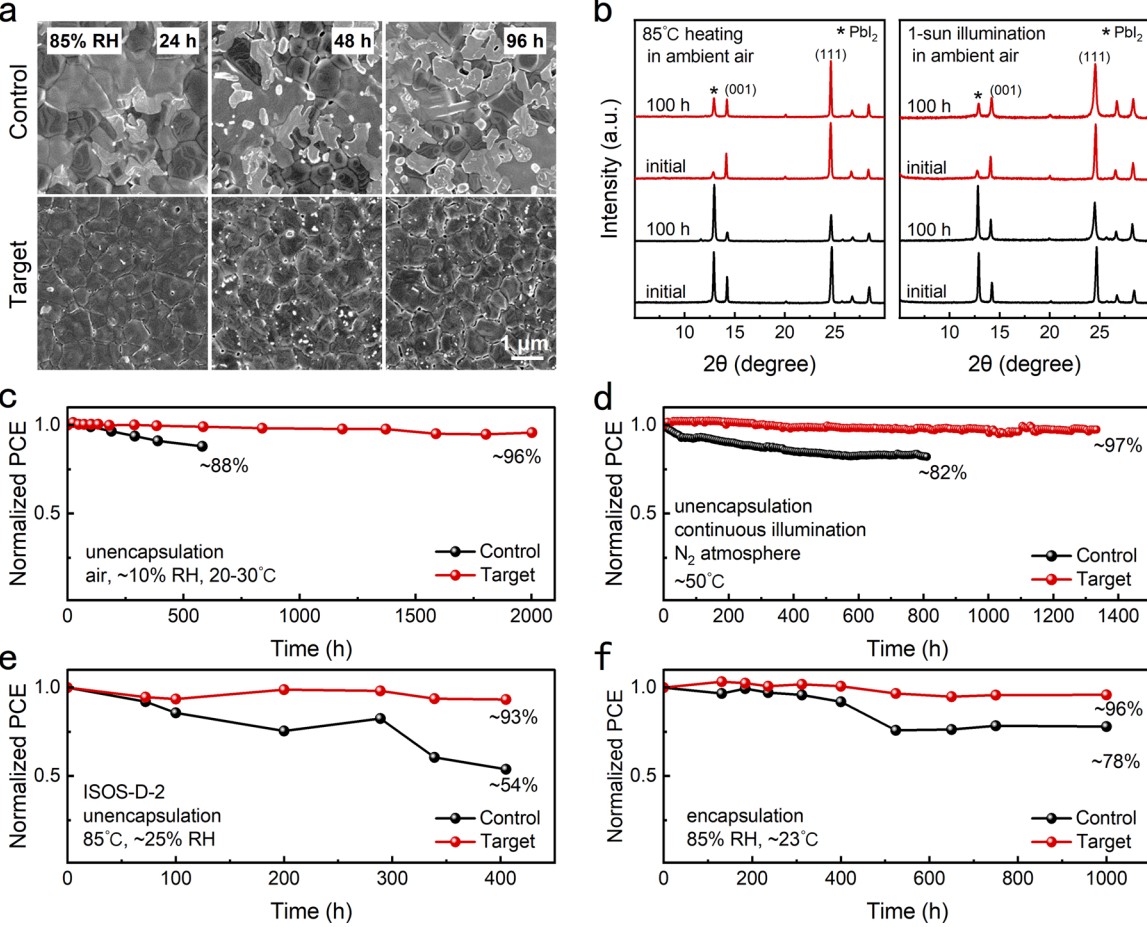

**Fig. 6 | Stability of perovskite films and device stability. a** Surface SEM images of films after storage in 85% RH at room temperature (-23°C). **b** XRD spectra of control and target films under 85°C heating (-25% RH) and 1-sun illumination in ambient air (20–30 °C, -25% RH), (**c**) Long-term stability of unencapsulated devices stored under ambient conditions of -10% RH and 20–30 °C. **d** Operational stability of the unencapsulated devices under continuous 1-sun illumination in N₂ glovebox. **e** PCE evolution of unencapsulated devices stored under -85 °C and -25% RH. **f** PCE evolution of encapsulated devices stored under -85% RH and -23 °C. Source data are provided as a Source Data file.

only 500 h (Fig. 6c). We also examined the long-term operational stability of unencapsulated devices under 1-sun illumination in the N₂ glovebox (Fig. 6d). The target device remains -97% of its initial PCE after 1300 h of operation. In comparison, the control device decreases to 82% of its initial PCE after 810 h of operation. The harsh conditions of high humidity and high temperature have been adopted to accelerate the device aging. We characterized the thermal stability of unencapsulated devices under heat condition at 85 °C and -25% RH according to the ISOS-D-2 protocol. As shown in Fig. 6e, the control device drops to 54% of the initial PCE after 400 h. By contrast, the target device just shows slight PCE decay. Before characterizing the stability of the device under high humidity of 85% RH, the device was roughly encapsulated. As shown in Fig. 6f, the PCE of the target device almost maintains 96% of its initial PCE, while the control device just maintains 78% of its initial PCE after aging for 1000 h. The initial PCEs of the above stability tests are shown in Supplementary Table 5. Taken together, the creeper-inspired strategy achieved by PHMG can effectively enhance PSCs' storage and operational stability, even under harsh conditions. When choosing a molecular to construct a molecular creeper, the molecular structure and function group should be carefully paid attention. Generally, a long-line or network molecular structure with soft characteristics can ensure the molecular creeper provide a reliable cover for the grain, and well-distributed function groups that interact with perovskite (FA⁺, I⁻, Pb²⁺) to serve as the suckers can anchor the perovskite component and make the creeper

to cover grain steadily. Preferably, the function group can interact with the FA⁺, since the FA⁺ as the organic component is ready to escape away, especially under external conditions.

## Discussion

In summary, we have proposed a creeper-inspired strategy by using the PHMG as the molecular creeper to inhibit perovskite decomposition, achieving high-efficiency PSCs with enhanced stability. The molecular creeper can strongly anchor FA⁺ through a strong interaction and steadily cover perovskite grain, which inhibits perovskite decomposition by suppressing organic component escape away. Owing to the obtained high-quality perovskite film, the corresponding PSCs exhibit a PCE of 25.42%, the highest among the TiO₂-based planar PSCs reported so far. Notably, both the perovskite film and device possess enhanced stability even under damp heat conditions. The device can retain -97% of its initial PCE after 1300 h of operation under 1-sun illumination. Our bio-inspired strategy also shows reliable applicability on the perovskite film fabricated by different methods, providing guidelines for solving bottleneck issues regarding the perovskite stability, and more opportunities to achieve efficient PSCs with excellent stability.

## Methods
### Materials
The patterned F-doped tin oxide glass (FTO, 7 Ω sq⁻¹) was purchased from Shangyang Solar Energy Technology CO., Ltd. (Suzhou, China).

Titanium tetrachloride (TiCl$_4$, 99.99%) precursor was purchased from Aladdin (Shanghai, China). Lead iodide (PbI$_2$, 99.999%) was purchased from Tokyo Chemical Industry (TCI). Rubidium chloride (RbCl) was purchased from Sigma Aldrich. Polyhexamethyleneguanidine hydrochloride (PHMG, ≥99%, MW = 533.032) was purchased from Macklin. $N$, $N$- dimethylformamide (DMF), dimethyl sulfoxide (DMSO), acetonitrile (ACN), chlorobenzene (CB), and isopropanol (IPA) were purchased from Sigma-Aldrich. Formamidinium iodide (FAI), methylammonium chloride (MACl), CsI, 2,2′,7,7′-tetrakis ($N$, $N$-dipmethoxyphenylamine)-9,9′spirobifluorene (Spiro-OMeTAD), tert-butylpyridine (tBP), li-bis(trifluoromethanesulfonyl)imid (Li-TFSI), Poly[bis(4-phenyl)(2,4,6-trimethylphenyl)amine] (PTAA) were purchased from Xi'an Yuri Solar CO., Ltd.

### Device fabrication
The FTO substrates were ultrasonically cleaned in the sequences of detergent solution, deionized water, ethanol, and deionized water for 15 min, respectively. After plasma cleaning, the FTO substrate was put into TiO$_2$ solution at 70 °C for 40 min.

The perovskite films were fabricated by two-step deposition. The 1.5 M PbI$_2$ with 3–5% molar RbCl was dissolved in a mixed solvent of DMF/DMSO with a volume ratio of 9:1 and was spin-coated onto TiO$_2$ substrates at 1500 rpm. for 30 s followed by annealing at 70 °C for 1 min in N$_2$ glovebox. For the target PbI$_2$ precursor, different ratios of PHMG (the molar ratio of PHMG/PbI$_2$ 0.25%, 0.5%, 1%) were introduced. Subsequently, the FAI: MACl (90:13 mg/ml) was dissolved in IPA and spin-coated onto the cooling PbI$_2$ films at 1800 rpm. for 30 s. Then, the films were annealed at 150 °C for 15 min under ambient air conditions of ~40% relative humidity.

The perovskite films were fabricated by one-step deposition with a precursor solution (Cs$_{0.05}$FA$_{0.85}$MA$_{0.1}$PbI$_3$, 1.54 M) with 4000 rpm. for 18 s and 800 µl diethyl ether as anti-solvent was dripped at 12 s before the end of spin coating. Then, the films were annealed at 100 °C for 10 min.

The MA-based perovskite films were fabricated by one-step deposition with a precursor (MAPbI$_3$ and MA$_{0.75}$FA$_{0.25}$PbI$_3$, both are 1.86 M) was spin-coated onto TiO$_2$ substrates at 4000 rpm. for 18 s, and 800 µl diethyl ether as anti-solvent was dripped at 12 s before the end of spin coating. Then, the films were pre-annealed at 60 °C for 1 min and annealed at 130 °C for 10 min.

The MeO-PEAI solution (16 mM) was spin-coated onto the FTO/TiO$_2$/perovskite surface at 4000 rpm. for 30 s without further annealing. The Spiro-OMeTAD (72.3 mg) with 26.6 µl tBP, and 18 µl Li-TFSI (520 mg/ml in ACN) was in 1 ml CB, which was spin-coated at 4000 rpm. for 30 s. The PTAA precursor consists of 30 mg PTAA, 6 µl Li-TFSI (520 mg/ml in ACN), 9 µl tBP, and 1 ml CB. The PTAA was deposited on the passivation layer at 3000 rpm for 30 s and then annealed at 100 °C for 5 min. When we carried out the operational tests and the tests under the condition of high temperature and high humidity, the Spiro-OMeTAD was changed to PTAA.

Au electrode (60 nm) was deposited by thermal evaporation on the obtained PSCs.

### Characterization
The interaction of FAI and PHMG is proved by 1H nuclear magnetic resonance (1H NMR, Ascend 600), Raman spectra (Horiba LabRAM HR Evolution), and Fourier transform infrared spectroscopy (FTIR, Bruker). The surface and cross morphologies of perovskite films were measured using cold field-emission scanning electron microscopy (SEM, Hitachi S-4800). The X-ray diffractometer (XRD, D8 Advance, Bruker) obtained crystalline perovskite films. PL mapping was carried out using a laser confocal fluorescence lifetime imaging microscope (Nikon-ARsiMP-LSM-Kit-Legend Elite-USX) with a 405 nm excitation wavelength. The signal collection area for PL mapping was 30 × 30 µm$^2$. The carrier behavior of perovskite films

was obtained by time-resolved photoluminescence and photoluminescence quantum efficiency (TRPL, PLQE, Edinburgh Instrument, FLS980). The excitation wavelength is 470 nm. The TRPL fitted curves were obtained according to the following biexponential equation:

Device efficiency was measured by using a Keithley 2400 source meter with a scanning rate of 0.06 V/s under simulated AM 1.5 G illumination and the mask area is 0.08 cm$^2$. The external quantum efficiency and integrating current of perovskite solar cells were measured by QE-R systems (Enli Tech). The unpackaged devices were stored in ambient air at room temperature (20–30 °C, ~10% RH) to measure the long-term aging stability. The operational stability tracking of the device was tested under light-emitting diode illumination (AM 1.5 G) at 50 ± 5 °C in an N$_2$ glovebox. The EQE spectra were measured via QE-R systems (Enli Tech.). PL-V hysteresis measurement used the electrochemical workstation (Zahner Zennium). Impedance spectra were measured using HIOKI impedance analyzer (IM 3570). Time-of-flight secondary ion mass spectrometry (TOF-SIMS 5-100, IONTOF GmbH) was performed to characterize the ionic migration of FA$^+$. Transient ion-migration current was obtained by electrochemical workstation (CHI660E).

### Density function theory simulations
All calculations used the Vienna ab initio simulation package (VASP)[52], and the exchange-correlation energy has been treated using the Perdew-Burke-Ernzerhof (PBE) function[53]. Van der Waals forces were taken into account in the calculations[54]. The cutoff energy was set up at 500 eV and the energy and force convergence parameters were set to 10$^{-5}$ eV and 0.05 eV/Å, respectively. The FAPbI$_3$ surface models were constructed from 3 × 3 × 1 supercells of α-FAPbI$_3$ (100). To avoid interactions between adjacent slabs, a vacuum layer of more than 20 Å was added to the surface in the 2D slab model. The formation energy of an FA vacancy was calculated by $E_f = E_{defect} - E_{perfect} - \mu_{FA} + q_{EF}$, where $E_{defect}$ and $E_{perfect}$ were the total energy of the α-FAPbI$_3$ (100) surface with and without an FA vacancy, respectively, $\mu_{FA}$ was the chemical potential of FA, q was the charge state of the FA vacancy (which was 0) and $E_F$ was the Fermi level of the perovskite. The absorption energy between the organic molecule and the surface of perovskite was calculated by $E_a = E_{molecule@perovskite} - E_{molecule} - E_{perovskite}$, where $E_{structure}$ is the total energy of the corresponding structure.

### Reporting summary
Further information on research design is available in the Nature Research Reporting Summary linked to this article.

## Data availability
All data generated in this study are provided in the article and Supplementary Information, and the raw data supporting this study are available from the Source Data file. Source data are provided with this paper.

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

## Acknowledgements

This work is supported partially by the Key Research and Development Program sponsored by the Ministry of Science and Technology (MOST) (Grant nos. 2022YFB4200301), National Natural Science Foundation of China (Grant nos. 52232008, 51972110, 52102245, and 52072121), Beijing Natural Science Foundation (2222076 and 2222077), Beijing Nova Program (20220484016), Young Elite Scientists Sponsorship Program by CAST (2022QNRC001), 2022 Strategic Research Key Project of Science and Technology Commission of the Ministry of Education, Huaneng Group Headquarters Science and Technology Project (HNKJ20-H88), the Fundamental Research Funds for the Central Universities (2022MS029, 2022MS02, 2022MS031, 2023MS042, and 2023MS047) and the NCEPU "Double First-Class" Program.

## Author contributions

M.L., S.D., and H.H. conceived the idea. M.L., P.C., and H.H. guided the work as supervisors. S.D. and H.H. did experimental designs, device fabrication, and data analysis. Z.L. participated in the device fabrication and some characterizations. M.W. conducted the DFT calculation and analysis. S.Q. and L.L. assisted with the in situ UV-vis, PL spectra, and PL-V hysteresis test. P.C. and C.S. contributed to the ion migration and defect-related characterization. S.D. and H.H. wrote the first draft of the manuscript. L.Y., Y.Y., and X.W assisted in revision and polishing the manuscript language. All authors discussed the results and contributed to the revisions of the manuscript.

## Competing interests

The authors declare no competing interests.
