## [Peer Review File · Nature Communications]

Inhibiting perovskite decomposition by a creeper-inspired strategy enables efficient and stable perovskite solar cellsREVIEWER COMMENTS

Reviewer #1 (Remarks to the Author):

The paper presents a new and subtle approach inspired from nature creeper to inhibit perovskite decomposition and enhance device stability. The author focuses on the organic component, the relatively unstable materials, and proposes that the perovskite stability should be addressed at an early stage, the annealing process. A molecular creeper was constructed to cover perovskite grain by rationally using PHMG, which inhibits perovskite decomposition from the annealing process. In addition, I think the experiment design and argument in this manuscript is well and adequate. The perovskite crystallization was investigated by in situ measurement, a novel measurement by analyzing PL hysteresis is designed to characterize ion migration, and various experiments was carried out to demonstrate the enhanced stability for film and device. Notably, a high-efficiency PSCs with a certificated PCE of 25.36%, a highest value among the TiO₂-based PSCs reported to date, was also obtained. It is really important to enhance the stability on an efficient PSCs. I believe that this work is of high quality and meaningful, which is deserved to be published after addressing the points listed below.

- (1) In the abstract, the in-situ spectrum measurements should directly mention the specific characterization.
- (2) The last sentence of second paragraph in introduction need to be modified, the main point should focus on the stability of film and device, the keynote of the whole manuscript.
- (3) The Ref. 25 can partially support the idea of perovskite decomposition is an accelerating process. however, if mentioned in main text, more proper and direct-supporting reference should be need.
- (4) The bio-inspired strategy can effectively enhance perovskite stability, since it can inhibit the FA⁺ migration and escape away, and eliminates the detrimental effect of PbI₂. Can this strategy be applied in other perovskite with different composition, especially the MA-based perovskite which suffers ion migration of MA⁺.
- (5) The manuscript mentioned that the PHMG can promote the reaction between PbI₂ and organic halide. What is the reason?
- (6) The J-V curves under forward scan should be supplemented.
- (7) Why is there a significant difference in the photoelectronic performance parameters of devices prepared by one-step and two-step methods? Please explain it. Additionally, please provide the cross-section SEM image of corresponding device so that we can know the thickness of the perovskite layer, which is a required parameter for SCLC testing (Supplementary Fig. 5).
- (8) Please provide the initial values for PCE in stability tests.

Reviewer #2 (Remarks to the Author):

This manuscript entitled Inhibiting perovskite decomposition by a creeper-inspired strategy enables efficient and stable perovskite solar cells addresses the urgent issues for promoting commercialization of PSCs, the device stability, and proposed a bio-inspired approach. Learning from the nature creeper, this manuscript constructs a molecular creeper by using PHMG to cover perovskite grain, providing protection for perovskite, which is a subtle and novel idea. Various results demonstrated that The PHMG can optimize perovskite crystallization, inhibit the FA⁺ migration and escape away, even under high temperature, inhibiting perovskite decomposition from the annealing process. Impressively, the resulting PSCs possess an PCE of 25.42% (certificated 25.36%), which is reported to be the highest among the TiO₂-based planar PSCs to date. Moreover, both the perovskite film and PSCs exhibit enhanced stability even under harsh condition. The investigation on the device stability from a subtle creeper-inspired perspective, especially the emphasis of dynamic stability of perovskite film during the annealing process and ion migration, should be meaningful to researchers who are interested in the topic of device stability. I suggest to publish the manuscript in Nature Communications after the following comments have been well addressed

(1) It would be more meaningful to provide more discussion about molecular creepers, specifically, how to choose a molecule as molecular creeper.

(2) How the PHMG is used when fabricating perovskite by two-step method? Whether the PHMG is incorporated into PbI₂ solution or organic salt solution? If it is PbI₂ solution, more characterization focusing on PbI₂ film should be supplemented. And the part of device fabrication should methon the detail method of PHMG application.

(3) The SEM images in larger range should be supplemented.

(4) AFM is suggested to be used to characterize perovskite film.

(5) How about the hysteresis of the PSCs? The relevant information and analysis should be supplemented. The J-V curves scanned in forward direction should be supplemented.

(6) How the PHMG promote the reaction between PbI₂ and organic halide?

(7) What is the average PCE for PSCs in Fig. 4d?

(8) There are minor grammar mistakes need to be addressed.

Reviewer #3 (Remarks to the Author):

In the manuscript, Xing et al. reported a strategy for repairing humidity-induced interfacial degradation in low-dimensional PSCs with expanded humidity printing window. The authors utilized adsorbed water as a nucleation site for atomic layer deposition, avoiding crystal damage while precisely eliminating degradation triggers, and the target device achieved impressive efficiency and stability. The authors provide a novel idea with detailed experimental characterization, which is instructive for upscaling fabrication of low-dimensional PSCs. However, some expressions and experimental contents in the work are still insufficient. Hence, the following points need be addressed before its acceptance:

1. The manuscript lacks a discussion of the current research on ACI PSCs. For the readers' convenience, an in-depth analysis of the existing optimization strategies for ACI PSCs is necessary.
2. The advantages of Al₂O₃ are reported in the manuscript, and the effects of other metal oxides by atomic layer deposition, such as SnO₂, etc., need to be added.
3. The description of residual H₂O_{ad} being susceptible to inducing crystal damage at elevated temperatures is puzzling, and the authors need to comment on it.
4. The authors claim that delayed appearance of InO₂ were observed in the aged reference sample, but it is difficult to observe this finding in Fig 4. The authors should provide the data with a longer sputtering time.
5. The statement in the manuscript needs to be adjusted for reading.

Response to Reviewers' Comments on the manuscript entitled “Inhibiting perovskite decomposition by a creeper-inspired strategy enables efficient and stable perovskite solar cells”

We appreciate the reviewers' valuable time and suggestions for our manuscript. According to your professional comments, we have carefully revised the manuscript. The point-by-point responses have also been prepared as follows. (The words in black are from Reviewers. Our responses are written in blue. The revised parts in the manuscript and supplemental information are marked in red.)

Reviewer #1 (Remarks to the Author):

The paper presents a new and subtle approach inspired from nature creeper to inhibit perovskite decomposition and enhance device stability. The author focuses on the organic component, the relatively unstable materials, and proposes that the perovskite stability should be addressed at an early stage, the annealing process. A molecular creeper was constructed to cover perovskite grain by rationally using PHMG, which inhibits perovskite decomposition from the annealing process. In addition, I think the experiment design and argument in this manuscript is well and adequate. The perovskite crystallization was investigated by in situ measurement, a novel measurement by analyzing PL hysteresis is designed to characterize ion migration, and various experiments was carried out to demonstrate the enhanced stability for film and device. Notably, a high-efficiency PSCs with a certificated PCE of 25.36%, a highest value among the TiO₂-based PSCs reported to date, was also obtained. It is really important to enhance the stability on an efficient PSCs. I believe that this work is of high quality and meaningful, which is deserved to be published after addressing the points listed below.

Response: We appreciate your recognition and recommendation of our work. We have carefully revised our manuscript according to your insightful comments one by one. We sincerely express our admiration for these valuable advices.

(1) In the abstract, the in-situ spectrum measurements should directly mention the specific characterization.

Response: Thanks for your constructive comments. The in situ spectrum measurements (in situ UV-vis spectra and in situ PL spectra) have played an important role in studying

the influences of PHMG on perovskite crystallization, and we have directly mentioned these measurements in the abstract according to your helpful suggestion.

The revised part in the abstract is as follows:

“The molecular creeper promotes the perovskite nucleation and delays the perovskite growth, which is demonstrated by in situ UV-vis and PL measurements”

(2) The last sentence of second paragraph in introduction need to be modified, the main point should focus on the stability of film and device, the keynote of the whole manuscript.

Response: Thanks for your valuable comments. We modify the last sentence of the second paragraph, emphasizing that the perovskite component integrity is important for the stability of perovskite film and device.

The supplemented sentence in the last of the second paragraph is as below:

“Hence, it is expected to be a feasible approach that managing the perovskite components and modifying perovskite film properties to enhance the film stability.”

(3) The Ref. 25 can partially support the idea of perovskite decomposition is an accelerating process. however, if mentioned in main text, more proper and direct-supporting reference should be need.

Response: Thanks for your insightful comments. Regarding the decomposition process of perovskite films and the corresponding PSCs, various reports have directly or indirectly demonstrated that it is an accelerating process, which can be seen from the evolution of film property and PSC performance. On the one hand, the perovskite decomposition is accompanied by the appearance of PbI_2 and I_2 , the former is light-unstable and the latter has high redox activity, which will further accelerate the decomposition of perovskite films. On the other hand, the decomposition of perovskite films will lead to the generation of defects and the accumulation of carriers at grain boundaries or interfaces, which can further accelerate the decomposition process.

According to your suggestion, we have added proper literature.

The supplemented literatures are as below:

[1] Wang S, et al. Accelerated degradation of methylammonium lead iodide perovskites induced by exposure to iodine vapour. Nat. Energy 2, 16195 (2016).

[2] Meng Y, et al. *Chemical Reaction Kinetics of the Decomposition of low-Bandgap tin–lead halide perovskite films and the effect on the ambipolar diffusion length.* *ACS Energy Lett.* **8**, 1699-1696 (2023).

[3] Zhou J, et al. *Modulation of perovskite degradation with multiple-barrier for light-heat stable perovskite solar cells.* *Nat. Commun.* **14**, 6120 (2023).

(4) The bio-inspired strategy can effectively enhance perovskite stability, since it can inhibit the FA^+ migration and escape away, and eliminates the detrimental effect of PbI_2 . Can this strategy be applied in other perovskite with different composition, especially the MA-based perovskite which suffers ion migration of MA^+ .

Response: We are grateful for your valuable suggestions. To explore the influence of PHMG in MA-based perovskite, we introduce the PHMG into the MAPbI_3 and $\text{MA}_{0.75}\text{FA}_{0.25}\text{PbI}_3$ precursor, respectively, and then fabricate the corresponding PSCs by one-step spin-coating methods. Subsequently, we evaluate the ion migration concentration through the transient ion-migration currents. As shown in Fig. R1 and Tables R1, R2, the mobile ion concentration of both PSCs based on MAPbI_3 and $\text{MA}_{0.75}\text{FA}_{0.25}\text{PbI}_3$ is decreased after incorporating PHMG, demonstrating that the creeper-inspired strategy can also be well applied in the MA-based perovskite.

The experimental data are supplemented in SI as Supplementary Fig. 16 and Supplementary Tables 3 and 4.

The revised part in the manuscripts:

“The effectiveness of creeper-inspired strategy on inhibiting ion migration has been also validated by the decreased mobile ion migration in both PSCs based on MAPbI_3 and $\text{MA}_{0.75}\text{FA}_{0.25}\text{PbI}_3$, indicating that this strategy can also be well applied in perovskite with different component (Supplementary Fig. 16 and Supplementary Tables 3 and 4).”

Fig. R1 Transient ion-migration currents and mobile ion concentrations of fresh (a) MAPbI_3 (b) $\text{MA}_{0.75}\text{FA}_{0.25}\text{PbI}_3$ devices. 1.1 V forward bias applied to the devices is removed. Five devices were tested in each group.

Table R1. The mobile ion concentration of MAPbI₃-PSCs without and with PHMG.

Sample	M				
W/O PHMG	15.8	17.4	19.3	11.7	13.1
With PHMG	3.83	3.81	4.09	3.85	5.65

Table R2. The mobile ion concentration of MA_{0.75}FA_{0.25}PbI₃-PSCs without and with PHMG.

Sample	M				
W/O PHMG	2.83	3.29	2.97	3.03	3.54
With PHMG	1.12	1.57	1.48	1.90	1.85

(5) The manuscript mentioned that the PHMG can promote the reaction between PbI₂ and organic halide. What is the reason?

Response: We appreciate your insightful question. In our experiment, the PHMG molecule is introduced into the precursor solution of PbI₂. We can find that the PbI₂ film has a smaller grain after incorporating the PHMG through SEM and AFM measurement (Fig. R2). The smaller grains are conducive to the full reaction between PbI₂ with organic salt. Hence, the reason why the PHMG can promote the reaction between PbI₂ and organic halide is that the PHMG reduces the grain size of PbI₂ films.

The experimental data are supplemented in SI as Supplementary Fig. 5.

The revised part is in the SI as the Supplementary Note 2:

“After incorporating PHMG into PbI₂ precursor, it can be seen the PbI₂ film has a smaller grain through measurements of SEM and AFM. The smaller grains are conducive to the full reaction between PbI₂ with organic salt.”

Fig. R2 (a) Surface SEM images and (b) AFM images of control and target PbI_2 films.

(6) The J - V curves under forward scan should be supplemented.

Response: Thanks for your valuable suggestion. We supplemented the forward scan curves of champion PSCs fabricated by two-step spin-coating and one-step spin-coating, respectively. The J - V curves and detailed photovoltaic parameters are also shown in Fig. R3.

The supplemented information has been added in Figs 4e, h in the manuscript.

Fig. R3 The J - V curves (forward and reverse scan) of control and target PSCs. (a) two-step spin-coating and (b) one-step spin-coating.

(7) Why is there a significant difference in the photoelectronic performance parameters of devices prepared by one-step and two-step methods? Please explain it. Additionally, please provide the cross-section SEM image of corresponding device so that we can know the thickness of the perovskite layer, which is a required parameter for SCLC testing (Supplementary Fig. 5).

Response: We appreciate your meticulous inquiry and helpful suggestion. As for the performance difference between PSCs prepared by one-step and two-step methods, there should be three reasons: Firstly, one-step and two-step methods have different modes of film formation. In the one-step method, the precursor solution was prepared strictly according to the ratio of perovskite, and the excess solvent was extracted by using anti-solvent. In the two-step method, PbI_2 film is preferentially prepared, and then organic ammonium reacts with PbI_2 . Secondly, the perovskite component and the corresponding stoichiometric ratio used in one-step and two-step methods are different. The perovskite used in the two-step method is $\text{FAPbI}_3\text{-RbCl}$, and the perovskite used in the one-step method is $\text{Cs}_{0.05}\text{FA}_{0.85}\text{MA}_{0.1}\text{PbI}_3$. Thirdly, the fabrication technology using the one-step method and two-step method in our group also has differences such as the annealing temperature, the spinning speed, the annealing time, and so on, which can influence the properties of perovskite film and further influence the PSCs performance.

In our manuscript, The PCEs for PSCs based on two-step and one-step methods are 25.42% and 25.41%, respectively. The PCE values are closed. Detailedly, the J_{SC} of PSCs based on two-step is slightly higher than that of PSCs based on one-step, and the V_{OC} of PSCs based on two-step is slightly lower than that of PSCs based on one-step. The reasons for these differences in J_{SC} and V_{OC} are associated with the three points we mentioned above, where the second one, the difference in the perovskite component and corresponding stoichiometric ratio should be the main reason.

The Cross-sectional SEM images of control and target perovskite films fabricated by the two-step method were supplemented (Fig. R4). The thickness of the perovskite films is ~ 800 nm, which supports the calculation of the trap density (N_{trap}) in SCLC testing.

We added Fig. R4 in SI as Supplementary Fig. 9.

Fig. R4 Cross-sectional SEM images of control and target perovskite films fabricated by two-step spin-coating. The thickness (L) of the perovskite films is ~ 800 nm.

(8) Please provide the initial values for PCE in stability tests.

Response: We appreciate your detailed inquiry. In Fig. 6c, for the test of long-term stability under ambient conditions of ~10% RH and 20-30°C, the initial PCEs for control and target PSCs are 23.83 and 24.41%, respectively. In Fig. 6d, for the test of long-term stability under continuous 1-sun illumination in the N₂ glovebox, the initial PCEs for control and target PSCs are 19.26% and 20.48%, respectively. In Fig. 6e, for the test of long-term stability under ~85°C and ~25% RH, the initial PCEs for control and target PSCs are 19.46% and 20.73%, respectively. In Fig. 6f, for the test of long-term stability under ~85% RH and ~23°C, the initial PCEs for control and target PSCs are 19.92% and 20.50%, respectively. When we carried out the operational tests and the tests under the condition of high temperature and high humidity, the Spiro-OMeTAD was changed to PTAA, which has been explained in the parts of methods. We have added the initial PCE value in Supplementary Table 5.

Reviewer #2 (Remarks to the Author):

This manuscript entitled Inhibiting perovskite decomposition by a creeper-inspired strategy enables efficient and stable perovskite solar cells addresses the urgent issues for promoting commercialization of PSCs, the device stability, and proposed a bio-inspired approach. Learning from the nature creeper, this manuscript constructs a molecular creeper by using PHMG to cover perovskite grain, providing protection for perovskite, which is a subtle and novel idea. Various results demonstrated that The PHMG can optimize perovskite crystallization, inhibit the FA⁺ migration and escape away, even under high temperature, inhibiting perovskite decomposition from the annealing process. Impressively, the resulting PSCs possess an PCE of 25.42% (certificated 25.36%), which is reported to be the highest among the TiO₂-based planar PSCs to date. Moreover, both the perovskite film and PSCs exhibit enhanced stability even under harsh condition. The investigation on the device stability from a subtle creeper-inspired perspective, especially the emphasis of dynamic stability of perovskite film during the annealing process and ion migration, should be meaningful to researchers who are interested in the topic of device stability. I suggest to publish the manuscript in Nature Communications after the following comments have been well addressed

Response: Thanks for your recognition and recommendation of our work. We have carefully read your suggestions, and have replied to the questions point-by-point. The appropriate revisions and supplements have been made, thank you again for your comments.

(1) It would be more meaningful to provide more discussion about molecular creepers, specifically, how to choose a molecule as a molecular creeper.

Response: We thank you for your valuable comments, providing a discussion about the molecular creepers is expected to provide more reference value for the readers. We noticed that the surface plays an important role in perovskite film stability. To anchor and maintain the component from escape away should be an effective approach to enhance the stability of perovskite film. We observe that the creepers can provide a reliable cover for the naked walls through strong suckers, which can decorate the wall, and further inhibits the components of the wall from escaping, even on rainy days with strong wind. If we imagine the perovskite grain as a house, we hope some molecules cover the grain surface, just like creepers cover the walls. Based on the above thinking,

we hope the molecule served as a creeper should have two points. Firstly, the molecular creeper should possess a long-line or network molecular structure with soft characteristics because this structure can ensure the molecular creeper provide a reliable cover for the grain; the second is that the molecular creeper should have function groups that can interact with perovskite (FA^+ , I^- , Pb^{2+}) to serve as the suckers, these group can anchor the component of perovskite and make the creeper to cover grain steadily. Preferably, we further hope the function group can distributed well in the long-line or network molecular. More importantly, the function group can interact with the organic component of FA^+ , since the FA^+ is ready to escape away, especially under external harsh conditions.

In our work, we utilize the PHMG to serve as a molecular creeper on perovskite to inhibit its decomposition. The PHMG possesses a long-line molecular structure where the guanidinium groups (distributed well in the PHMG molecule) can serve as suckers that strongly anchor FA^+ through multiple hydrogen bonds.

The revised part is in the last paragraph of the revised manuscripts:

“When choosing a molecular to construct a molecular creeper, the molecular structure and function group should be carefully paid attention. Generally, a long-line or network molecular structure with soft characteristics can ensure the molecular creeper provide a reliable cover for the grain, and well-distributed function groups that interact with perovskite (FA^+ , I^- , Pb^{2+}) to serve as the suckers can anchor the perovskite component and make the creeper to cover grain steadily. Preferably, the function group can interact with the FA^+ , since the FA^+ as the organic component is ready to escape away, especially under external conditions.”

(2) How the PHMG is used when fabricating perovskite by two-step method? Whether the PHMG is incorporated into PbI_2 solution or organic salt solution? If it is PbI_2 solution, more characterization focusing on PbI_2 film should be supplemented. And the part of device fabrication should mention the detail method of PHMG application.

Response: We sincerely appreciate your important suggestion regarding the experiment part. The PHMG is introduced into the PbI_2 precursor when fabricating perovskite by two-step method. We have supplemented the measurement of SEM and AFM to characterize the PbI_2 film. As shown in Fig. R5, both the results of SEM and AFM proved that the PbI_2 grain became smaller. In our manuscript, we observe that the introduced PHMG can promote the reaction between PbI_2 and organic halide, leading

to an accelerated nucleation. Based on the characterization of PbI_2 , the smaller grains should be the reason for the promoted reaction between PbI_2 and organic halide.

The experimental data are supplemented in SI as Supplementary Fig. 5.

The revised part is in the SI as the Supplementary Note 2:

“After incorporating PHMG into PbI_2 precursor, it can be seen the PbI_2 film has a smaller grain through measurements of SEM and AFM. The smaller grains are conducive to the full reaction between PbI_2 with organic salt.”

We supplemented the experiment details in the part of the method in the revised manuscript.

The revised part is as below:

“For target PbI_2 precursor, different ratios of PHMG (the molar ratio of PHMG/PbI_2 0.25%, 0.5%, 1%) was introduced.”

Fig. R5 (a) Surface SEM images and (b) AFM images of control and target PbI_2 films.

(3) The SEM images in larger range should be supplemented.

Response: Thanks for your suggestion. To more comprehensively and intuitively show the evolution of the PbI_2 content and the perovskite surface morphology during the annealing, we supplemented the SEM images with a wide range of magnification (Fig. R6).

The experimental data are supplemented in SI as Supplementary Fig. 2.

Fig. R6 The surface SEM images of control and target perovskite films.

(4) AFM is suggested to be used to characterize perovskite film.

Response: Thanks for your valuable comment. Following your suggestion, we carried out KPFM measurement to characterize perovskite film after annealing. As shown in Fig. R7a, compared to the target film, although it is difficult to show the residual PbI_2 on the control film in the AFM image, we can still notice that there are many tiny particles located at grain boundary on the control film, which may correspond to the PbI_2 . From the image of surface potential shown in Fig. R7b, we can find that the control film possesses a lower surface potential, which should result from the residual PbI_2 . The results of KPFM can validate that the perovskite undergoes decomposition accompanied by the generation of PbI_2 in the annealing process, and the creeper-inspired strategy can effectively inhibit this decomposition.

The experimental data are supplemented in SI as Supplementary Fig. 3.

The discussion about the results of KPFM has been supplemented in Supplementary Note 1.

The revised part of the manuscript is as below:

“The results of Kelvin probe force microscopy (KPFM) also validate that the perovskite undergoes decompose accompanied by generation of PbI_2 in the annealing process, and the creeper-inspired strategy can effectively inhibit this decomposition (Supplementary Fig. 3 and Supplementary Note 1).”

Fig. R7 (a) The AFM images and (b) KPFM images of control and target perovskite films.

(5) How about the hysteresis of the PSCs? The relevant information and analysis should be supplemented. The J-V curves scanned in the forward direction should be supplemented.

Response: Thanks for your meticulous suggestion. We supplemented the forward scan curves of champion PSCs fabricated by two-step spin-coating and one-step spin-coating, respectively. The J-V curves and detailed photovoltaic parameters are shown in Fig. R8.

We calculate the hysteresis index (HI) of PSCs according to the following formula:

$$HI = \frac{PCE_{\text{Reverse scan}} - PCE_{\text{Forward scan}}}{PCE_{\text{Reverse scan}}} \times 100\%$$

As for the PSCs fabricated by the two-step method, the hysteresis of control and target PSCs are 2.3% and 1.0%, respectively. As for the PSCs fabricated by the one-step method, the hysteresis of control and target PSCs are 2.2% and 0.7%, respectively.

In general, the hysteresis effect in PSCs is associated with ion accumulation and carrier accumulation. Although all the PSCs show a small hysteresis, the target PSCs still show a smaller hysteresis than control PSCs, which should result from the mitigated ion migration and reduced defect states.

The modified figure is exhibited in Fig. 4e, h.

The revised part of the manuscript is as below:

“The target PSCs achieve a champion PCE of 25.42% ($J_{sc}=26.20 \text{ mA cm}^{-2}$, $V_{oc}=1.161 \text{ V}$, $FF=83.53\%$) with a minor hysteresis (Fig. 4e).”

Fig. R8 The J - V curves (forward and reverse scan) of control and target PSCs. (a) two-step spin-coating and (b) one-step spin-coating.

(6) How the PHMG promote the reaction between PbI_2 and organic halide?

Response: Thanks for your meticulous question. In the two-step sequential deposition method, the property of PbI_2 film has a huge influence on the reaction kinetics between PbI_2 and organic salt. We carried out measurements of SEM and AFM to characterize the PbI_2 film and found that the PbI_2 grain can become smaller after incorporating PHMG (Fig. R9). The smaller grains are conducive to the full reaction between PbI_2 with organic salt. Hence, the reason why the PHMG can promote the reaction between PbI_2 and organic halide is that the PHMG reduces the grain size of the PbI_2 film.

The experimental data are supplemented in SI as Supplementary Fig. 5.

The revised part is in the SI as the Supplementary Note 2:

“After incorporating PHMG into PbI_2 precursor, it can be seen the PbI_2 film has a smaller grain through measurements of SEM and AFM. The smaller grains are conducive to the full reaction between PbI_2 with organic salt.”

Fig. R9 (a) Surface SEM images and (b) AFM images of control and target PbI_2 films.

(7) What is the average PCE for PSCs in Fig. 4d?

Response: We are appreciative of your inquiry. The average PCEs of control and target PSCs in Fig. 4d are 23.80% and 24.84%, respectively.

The revised part of the manuscript is as below:

“The PCE distribution histograms indicate that both the PSCs possess good reproducibility and the target PSCs possess a higher average PCE of 24.84% than that (23.80%) of control PSCs (Fig. 4d).”

(8) There are minor grammar mistakes need to be addressed.

Response: We appreciate your helpful suggestion. We have carefully examined the full manuscript and made appropriate revisions. Thanks again.

Reviewer #3 (Remarks to the Author):

Comments on “Inhibiting perovskite decomposition by a creeper-inspired strategy enables efficient and stable perovskite solar cells”

In the manuscript by Du et al., the authors demonstrated a creeper-inspired strategy to enhance the efficiency and stability of PSCs. They proposed adopting PHMG as a molecular creeper on perovskite to fix the FA cation, which promoted the perovskite crystallization and inhibits its decomposition, confirmed by in situ UV-vis spectra, in situ PL spectra, TRPL, and PL-V hysteresis test. Therefore, the resulting TiO₂-based PSCs achieved a state-of-the-art PCE of 25.42% (certified 25.36%) with an operational stability of over 1300 hours. It might be interesting for the researchers and manufacturers to fabricate high-performance PSCs. However, this manuscript lacks of scientific integrity, novelty, data consistency, and reliability. This manuscript lacks of scientific integrity and novelty. The authors utilized a polyexamethyleneguanidine hydrochloride to modify the perovskite crystallization by using guanidinium groups. However, there are a lot of previous works based on additives with guanidinium groups to improve the perovskite crystallization, and the anchor ability of guanidinium groups with FA cations has been already reported (Nano Lett., 2016, 16, 1009-1016; Nat. Energy, 2017, 2, 972-979; Science, 2018, 360, 1442-1446; Nat. Commun., 2019, 10, 3008; Science, 2019, 364, 475-479; Solar RRL, 2021, 5, 2100097; Chem. Eng. J., 2022, 437, 135181; Mater. Chem. Front., 2023, 7, 2507-2527).

Response: We thank you very much for reviewing this manuscript and giving considerable inquiry. According to your comments, we have carefully revised the manuscript and replied to the questions point-by-point.

We appreciate your recognition of our manuscripts and think it is interesting for researchers and manufacturers to fabricate high-performance PSCs. We have rethought the novelty and highlight of our work based on the literature research, including the paper you mentioned. The novelty of our work is that we proposed a creeper-inspired strategy to construct a molecular creeper to cover the perovskite grain steadily, which has obviously enhanced the stability of perovskite film starting from the annealing process. Since perovskite decomposition is an accelerating process, the best time to address decomposition is the initial stage, the annealing process.

Our original design regarding the creeper-inspired strategy is that the creeper can realize steady cover using its suckers, hence the molecule served as a creeper should

have two points. Firstly, the molecular creeper should possess a long-line or network molecular structure with soft characteristics because this structure can ensure the molecular creeper provide a reliable cover for the grain. Secondly, the molecular creeper should have function groups that can interact with perovskite (FA^+ , I^- , Pb^{2+}) to serve as the suckers, these groups can anchor the component of perovskite and make the creeper cover grain steadily. Preferably, we further hope the function group can interact with the organic component of FA^+ since the FA^+ is ready to escape away, especially under external harsh conditions. Based on the above analysis, we choose the new PHMG to serve as the molecular creeper because PHMG is a polymer that possesses a long-lined structure with well-distributed guanidinium groups, such a feature can ensure the PHMG provides a reliable cover on perovskite through strongly interacting with FA^+ .

As for the interaction between the guanidinium group and FA^+ , we have carefully read and analyzed the 8 papers mentioned by the reviewer, and we found that the main research on these papers is using the guanidine cation and its salt to passivate defects, enhance crystallization, form low-dimension perovskite, and so on. Moreover, to our knowledge, we have not found any published paper that focuses on the immobilizing of FA^+ using guanidinium groups.

As for the reported guanidinium groups, these 8 papers actually focus on the guanidine cation, which is different from PHMG, since PHMG is a polymer with a long-line structure where the guanidinium groups are inserted in the middle. We carefully compared the PHMG with the guanidinium materials used among the 8 papers. As shown in Figure R10, the PHMG is a polymer material and the guanidinium group is distributed in the middle, which is different from other guanidinium materials. In addition, the PHMG is a new material for fabricating PSCs. The guanidine cation has been reported to passivate defects, enhance crystallization, form low-dimension perovskite, and so on. In our work, the PHMG is used to serve as a molecular creeper to cover perovskite grain, its function on perovskite is inspired by the natural creeper, showing a big difference from the reported works. Indeed, the new material of PHMG is mainly utilized to construct molecular creeper on perovskite grain, the influence of PHMG on perovskite crystallization is an additional scientific point, which distinguishes our work from that focus on perovskite crystallization, including utilizing guanidinium groups.

In short, our work emphasized the perovskite stability issue from the annealing process, and then a creeper-inspired strategy was proposed to effectively enhance the perovskite film stability, leading to efficient PSCs with high operational stability. From the scientific idea and new materials utilization, our work can be distinguished from reported works and provide guidelines for solving bottleneck issues regarding perovskite stability.

We added the original design regarding the creeper-inspired strategy, especially the selection of material for molecular creeper in the last paragraph of the revised manuscripts:

“When choosing a molecular to construct a molecular creeper, we molecular structure and function group should be carefully paid attention. Generally, a long-line or network molecular structure with soft characteristic can ensure the molecular creeper provide a reliable cover for the grain, and well-distributed function groups that interact with perovskite (FA^+ , I , Pb^{2+}) to serve as the suckers can anchor the perovskite component and make the creeper to cover grain steadily. Preferably, the function group can interact with the FA^+ , since the FA^+ as the organic component is ready to escape away, especially under the external condition.”

Fig. R10 The comparison of molecular structure between PHMG and guanidinium materials reported in reviewer mentioned works

Moreover, the author proposed that the perovskite stability results from the organic cation escape **during** the annealing process. Based on this, the authors utilized PHMG to stabilize FA cations and cover the perovskite. However, if the stability issues are simply caused by the escape of FA cations **during** the annealing process, researchers only need to increase the concentration of FAI in the perovskite precursor solution

appropriately to solve the problem, without the need for such a complex strategy, which further reduced the scientific integrity and novelty of this manuscript.

Response: We thank you for your valuable comments. We all know that, stability is a complex and urgent issue for PSCs, which can be induced by ion migration, component escape, phase separation, phase transition, and so on. These points need massive endeavors from different perspectives. Organic cation escape is a vital issue that induces perovskite decomposition. Gao's group observed a two-step perovskite decomposition process, i.e., the initial loss of cation followed by the collapse of perovskite structure at the atomic scale. (*Nat. Commun.*, 12, 5516 (2021)). This cation escape can occur starting from the annealing process, and throughout the entire process of decomposition. In our original abstract, we mentioned that we adopt PHMG as a molecular creeper on perovskite to inhibit its decomposition “**from**” the annealing process, which means the creeper-inspired strategy can enhance perovskite stability throughout its entire process, not only “**during**” annealing process.

Increasing the concentration of FAI in the precursor may have a positive effect on compensating the escape FA^+ and reducing the residual PbI_2 during annealing process. However, the changes in stoichiometric ratios certainly show a detrimental impact on the PSCs' efficiency, and even device stability through defects induced by non-stoichiometry and severe I^- accumulation (*Energy Environ. Sci.* 13, 258-267 (2020), *Nat. Commun.* 12, 6955 (2021), *Adv. Mater.* 35, 2211742 (2023)). Moreover, the perovskite stability after annealing still needs to be carefully addressed, especially during the PSCs operation. In our work, we proposed a creeper-inspired strategy to protect the perovskite, which can continuously inhibit its decomposition throughout its whole life. This creeper-inspired strategy successfully enhanced the stability of perovskite film and PSCs, even under the damp-heat conditions.

The data of this manuscript is lack of consistency and reliability. There are three typical examples.

Response: Thanks for your valuable time and comments. We carefully checked the data in the manuscript and found that all the data are reliable and steadily support the scientific idea. We do not think you mentioned three typical examples can demonstrate that our data is lack of consistency and reliability. In spite of this, we have made appropriate responses and revisions one by one, according to your comments.

Firstly, in the supplementary Fig. 2, the target perovskite showed a decrease of peak intensity during the annealing, which is inconsistent with the statements on the Line 169-172 at Page 7.

Response: We are appreciative of your valuable comments about the XRD results. Following your comment, we have rechecked the original XRD data. We find that an inappropriate description has been made as you mentioned, which has been revised in this version. Actually, this description does not influence the conclusion that the construction of molecular creeper by PHMG can effectively inhibit the perovskite decomposition, including in the annealing process.

In detail, we observe that the (001) peak of the target film shows a slight decrease as the annealing time increases. When we analyzed the XRD results, we normalized the XRD data corresponding to the different annealing times, which is the reason for the inconsistent statements on Lines 169-172 on Page 7. The supplementary Fig. 4 shows the original XRD data, and we further show the evolution of the PbI₂ peak and PbI₂/(001) ratio in Fig. 2b, c according to the original XRD data without normalization. From the results of XRD, we can still observe that the control perovskite shows an obvious decomposition accompanied by an increase in the PbI₂ peak, while the decomposition can be effectively inhibited by using our creeper-inspired strategy. The results of XRD are well kept in agreement with the SEM results and supplemented AFM results, which collaboratively demonstrates that the creeper-inspired strategy can inhibit perovskite decomposition, including in the annealing process.

Secondly, in the Fig. 3 and statements on the Line 206-207 at Page 8, the author proposed the PHMG can regulate the perovskite nucleation, which is inconsistent with the statements of promotion of perovskite crystallization on the Line 215-217 at Page 9. Moreover, the nucleation process is a very microscopic process, with the critical nucleation size generally believed to only contain a few dozen atoms. The obvious PL signal indicates that it has already crystallized instead nucleated.

Response: We appreciate your meticulous comments. Directly and accurately characterizing the perovskite crystallization is important for achieving high-quality perovskite film and efficient PSCs. In our work, we directly prove that the PHMG can accelerate the perovskite nucleation and delay the crystal growth through the in situ UV-vis and PL spectra. According to the traditional crystallization theory,

crystallization theoretically includes two processes: nucleation and crystal growth, and these two processes cannot be completely separate and sometimes can coexist (*Materials Science and Engineering, written by William D Callister*).

In our manuscript, we prepared perovskite films using the two-step method and then carried out the in situ measurements to monitor the crystallization process (Fig. R11). Perovskite crystallization is a microscopic and complex process, and there is not a clear boundary between nucleation and crystal growth when describing the crystallization process. To accurately describe the crystallization process, generally speaking, the wet-film stage of perovskite films during preparation is dominated by nucleation, while the solid-film stage during annealing is dominated by crystal growth (*Angew.Chem. Int.Ed. n/a, e202319282 (2024), Adv. Mater. 33, 2105290 (2021)*).

Fig. R11 The schematic diagram of two-step spin-coating

On the Line of 206-207 on Page 8, when we described the perovskite crystallization after dropping organic salt to form a wet film, we proposed that “*the peak of PL spectra for the spin-coating process is located at ~ 780 nm, indicating the formation of α -phase FAPbI_3 . At this stage, the formed FAPbI_3 may be the initial nucleus for the following crystallization*”. During the wet-film process, we think the nucleation is dominant in this stage.

On lines 215-217 on Page 8, when we described the perovskite crystallization during annealing, we proposed that “*during the annealing process, we can deconvolute the crystallization into three stages combining the results of in situ UV-vis and PL. Stage I corresponds to the fast crystallization which can be evidenced by the rapid intensity increase at UV-vis and PL spectra. The following sharp intensity decrease should be due to the solvent volatilization which makes the wet film transfer to solid film.*” Stage I is the beginning stage of annealing, corresponding to the wet film transforming to the solid film due to solvent volatilization. We think this stage can coexist the nucleation and crystal growth and this stage is a quick crystallization process since the applied high temperature is the dominant factor influencing the film in this stage, and the PL

intensity evolution of both films is similar. At the later stages II and III, we can observe that the PHMG can delay the crystal growth from the in situ PL results.

Based on the above analysis, we think the statement about the spin-coating process and annealing process is not inconsistent. We admit that the crystallization process of perovskite is quite complex, and it is difficult to characterize the crystallization process in situ intuitively. In our work, we used in situ spectroscopy to demonstrate that PHMG accelerates nucleation during the spin-coating stage and delays crystal growth during the annealing stage. However, although we carried out in situ measurements, it is still difficult to accurately distinguish the different physical behaviors during the crystallization process on a fine time scale. We believe that with the continuous deepening of research and the advancement of characterization techniques, our understanding of perovskite crystallization will become more profound and comprehensive.

In addition, as the issue of whether the PL can characterize the nucleation process, there are reported works that utilize the in situ PL to research the perovskite crystallization process, including the nucleation. As shown in Fig. R12a, Shi et al. reported the oriented nucleation in formamidinium perovskite fabricated by two-step method. They show the evolution of the PL spectra during the nucleation stage of perovskite films to illustrate their point of the sluggish nucleation kinetics (*Nature* 620, 323-327 (2023)). As shown in Fig. R12b, Müller-Buschbaum et al. studied the crystallization behavior of organic-inorganic perovskites during the spin-coating process. They divided the evolution into four steps. In phase II, the anti-solvent drop triggers the immediate emergence of an intense and broad PL peak, indicating the PL emission arises from the instantly formed perovskite nanocrystals. Their GIWAXS data demonstrated that the metastable nucleated phase causes PL luminescence (*Nat. Commun.* 12, 5624 (2021)). It can be seen that the in situ PL measurement as an effective technique can be utilized to describe the perovskite crystallization process, including the nucleation.

To describe the crystallization more accurately, we added a detailed description of stage I of the in situ PL spectra during the annealing process:

“Stage I corresponds to the fast crystallization which can be evidenced by the rapid intensity increase at UV-vis and PL spectra. Notably, stage I should coexist the nucleation and crystal growth since the film begins to be heated.”

We also supplemented a concise description of the influence of PHMG on perovskite crystallization through these in situ measurements:

“Based on this in situ characterization, the PHMG which served as the molecular creeper can accelerate the nucleation and delay the crystal growth, which is beneficial for improving the property of perovskite film.”

Fig. R12 The evolution of the PL spectra during the nucleation stage of perovskite films. (a) in situ PL of perovskite nucleation without and with PAD and (b) the evolution of structural and optoelectronic phases recorded by GIWAXS and PL, and the phase II starts from 25 s due to the antisolvent dripping.

Thirdly, the performances of champion PSCs are not consistent in the supplementary Fig. 6 and Fig. 4e, which is hard to explain.

Response: We appreciate your detailed inquiry. We all know that the efficiency of PSCs depends on massive factors. Hence, it is rational to first determine a relative optimal concentration of additive and then continue the research and achieve the champion PCE. This technological process of the experiment research is also widely adopted in previous works (*Science* 377, 531–534 (2022); *Nat. Photonics* 13, 460–466 (2019); *Science* 372, 1327–1332 (2021); *Nat. Photon.* 17, 96–105 (2023); *Adv. Mater.* 35, 2211806 (2023); *Adv. Energy Mater.* 2304521 (2024); *Science* 382, 810–815 (2023)).

In our manuscript, the supplementary Fig. 6 exhibits the photovoltaic performance of PSCs with different concentrations in a preliminary experiment in which we want to decide a relative optimal concentration of PHMG. After determining the optimal concentration of PHMG, then we carried out a massive experiment and obtained the

best-performing PSCs with the champion PCE for both control and target PSCs. Fig. 4e exhibits the photovoltaic performance of best-performing PSCs with the champion PCE. In our manuscript, we propose that “*After determining the optimal concentration of PHMG (Supplementary Fig. 6)*” and “*The target PSCs achieve a champion PCE of 25.42% with J_{sc} of 26.20 mA cm⁻², V_{oc} of 1.161 V, and FF of 83.53% (Fig. 4e)*”.

Therefore, from the perspective of scientific integrity, novelty, data consistency and reliability, I did recommend the publication of this manuscript in Nature Communications.

Response: From the above response, we have made a discussion about the scientific point, the novelty design, data demonstration, and so on from different perspective. We believe this detailed and in-depth discussion is beneficial to polish the scientific novelty. We also believe that the above response illustrates that our work indeed possesses scientific integrity, novelty, data consistency, and reliability, especially after the improved exploration considering the comments. Thanks again.

REVIEWERS' COMMENTS

Reviewer #1 (Remarks to the Author):

The author has appropriately answered my questions. I suggest accepting it as it is.

Reviewer #2 (Remarks to the Author):

The author has addressed all my concerns and the quality of the manuscript has been improved. I thus recommend to publish in Nature Communications.

Reviewer #3 (Remarks to the Author):

In the revised manuscript, the authors did the hard work to response my concerns and revise the manuscript. Although the issues of inconsistency in several data sets were addressed using explanations like "Hence, it is rational to first determine a relative optimal concentration of additive and then continue the research and achieve the champion PCE," as well as emphasizing that it should be "from" instead of "during". It cannot be denied that the data consistency in this article is poor. Such manuscript, if published in high-impact journals like Nature Communications, could potentially mislead readers and impact the journal's reputation. Therefore, I cannot recommend the publication of this manuscript on Nature Communications.